# Bayesian Exploration of Phenomenological EoS of Neutron/Hybrid Stars with Recent Observations

Emanuel V. Chimanski [1,*,†] , Ronaldo V. Lobato [2,3,†] , Andre R. Goncalves [4,†] and Carlos A. Bertulani [2,†]

1   National Nuclear Data Center, Brookhaven National Laboratory, Upton, NY 11973, USA
2   Department of Physics and Astronomy, Texas A&M University, Commerce, TX 75428, USA
3   Departamento de Física, Universidad de los Andes, Cra 1E, 18A-10, Bogotá 111711, Colombia
4   Computer Engineering Directorate, Lawrence Livermore National Laboratory, Livermore, CA 94720, USA
*   Correspondence: chimanski@bnl.gov
†   These authors contributed equally to this work.

**Abstract:** The description of the stellar interior of compact stars remains as a big challenge for the nuclear astrophysics community. The consolidated knowledge is restricted to density regions around the saturation of hadronic matter $\rho_0 = 2.8 \times 10^{14}$ g cm$^{-3}$, regimes where our nuclear models are successfully applied. As one moves towards higher densities and extreme conditions up to the quark/gluons deconfinement, little can be said about the microphysics of the equation of state (EoS). Here, we employ a Markov Chain Monte Carlo (MCMC) strategy to access the variability at high density regions of polytropic piecewise models for neutron star (NS) EoS or possible hybrid stars, i.e., a NS with a small quark-matter core. With a fixed description of the hadronic matter for low density, below the nuclear saturation density, we explore a variety of models for the high density regimes leading to stellar masses near to 2.5 $M_\odot$, in accordance with the observations of massive pulsars. The models are constrained, including the observation of the merger of neutrons stars from VIRGO-LIGO and with the pulsar observed by *NICER*. In addition, we also discuss the possibility of the use of a Bayesian power regression model with heteroscedastic error. The set of EoS from the Laser Interferometer Gravitational-Wave Observatory (LIGO) was used as input and treated as the data set for the testing case.

**Keywords:** Bayesian inference; MCMC; equation of state; neutron star; astrophysics

## 1. Introduction

Neutron stars (NS) are supernova remnants with a strong gravitational field and rapid rotation. They are objects with nuclear matter in one of the highest density states in the universe. The matter in their interior is compacted to values from a few g cm$^{-3}$ on their surface to possibly more than $10^{15}$ g cm$^{-3}$ in their center. NS have become, alongside black holes, sources of gravitational waves, and although their existence has been known for more than 50 years as pulsars [1], their internal structure still is not thoroughly understood. Part of the challenge of modeling the internal structure of these objects is related to our limited knowledge of nuclear properties in extreme physical environments, e.g., ultra-high densities and temperatures.

Recently, this picture has started to change with multimessenger observations [2] from binary NS mergers [3,4]. The constraints coming from these observations have started to provide the opportunity for a more detailed study of some of the parameters that describe global properties of NS such as radius constraints [5,6], tidal deformability [7], maximum mass [8] and other global properties. All this information is intimately associated with the equation of state (EoS) of the NS, and once one learns more about global properties, the microphysics can be constrained and studied. The GW170817 event, for example, besides the breakthrough of being the first gravitational wave (GW) detection of a merger of two neutron stars, became the subject of many studies that considered the impact of

the observation on internal aspects of the NS. The impact of the NS crust on the equation of state was investigated [9], as well as the effects of an isovector–scalar meson into the quark–meson coupling description of nuclear matter [10], and also different Skyrme-like parametrization [11]. Nonparametric inference showed that the event favors soft EoS [12]. Critical examinations of the EoS of dense matter were performed [13] considering the nuclear physics in the chiral effective field theory framework, but still left some understanding to be improved in regions of high densities of the EoS. The association of the event with an electromagnetic (EM) counterpart led to the first joint GW-EM constraints on the NS EoS. Using the binary's tidal deformability parameter, which describes how much a body is deformed by tidal forces [6,14–16], simulations of EM observations within numerical relativity and Kilonova models were performed, and extreme EoS models were ruled out: the stiffest and softest ones, e.g., see Figure 2 of Ref. [2]. Statistical Bayesian methods were applied in the context of the GW170817 event, where microscopic models of cold neutron stars using chiral effective models [17] were studied. Recently, the GW event with X-ray sources were combined and studied with the relativistic mean field models [18]. Besides the electromagnetic counterpart of the binary merger, another important recent electromagnetic measurement was done by NASA's *Neutron Star Interior Composition Explorer*

(*NICER*) [19], constraining the mass–radius of the pulsar PSR J0030+0451 [20,21]. In this case, nonparametric inference showed that the stiffer EoS [22] are favored. While the astronomical data were gathered and studied theoretically, experiments on Earth also have been performed. For example, the Lead Radius EXperiment (PREX-2) which has provided a better understanding of the nuclear matter around the saturation density and has a direct implication for the neutron star crust. The extrapolation of the data to higher densities has limited the stellar radii to $13.25 \lesssim R_{1.4} \lesssim 14.26$ km, meaning that the EoS should have a softening in the intermediate region and a stiffening at the high densities. This, in turn, could lead to a phase transition in the stellar core. The increment in observational data has helped to establish further constraints on the dense matter EoS, opening a rich field for statistical and machine learning models [23–26].

The description of nuclear matter, around the nuclear saturation density $\rho_0 \simeq 2.8 \times 10^{14}$ g cm$^{-3}$, $n_0 \simeq 0.17$ fm$^{-3}$, is well understood in terms of hadron physics. The microphysics at intermediate densities, i.e., densities above the nuclear saturation density and below the very high density from perturbative Quantum Chromodynamics (pQCD), is yet far away from a consensus, with a wide range of possible models. The debate includes the binding nature of NS, with theories considering self-bound quarks or simply gravity-bound systems. The asymptotic behavior of the EoS, on the other hand, has been understood in the context of quark matter [27], where pQCD techniques become accurate. As the details of the nuclear models are out of the scope of this work, we refer to Refs. [28–35] and references within for more information.

In this work, we separate the description of the equation of state into three pieces, a fixed hadronic model for low densities up to the nuclear saturation density and two polytropic functional for the intermediate-higher densities up to the pQCD limit, $\rho_0 < \rho < \rho_{pQCD}$. We based our approach on the work by Read et al. 2009 [36], where a piecewise EoS was fitted with a direct cost function minimization. Here, considering this picture, we consider a larger class of models made possible by modern computing resources and with up-to-date observations. We adjust the density transition between the two polytropes, and then perform a Bayesian Inference with Markov Chain Monte Carlo (MCMC) on the adiabatic index of each polytrope. We also consider a case where the transition density is also inferred from MCMC. This approach provides an assessment of the impact of variations in the EoS at intermediate and high densities up to the pQCD limit on the mass radius diagram of the star.

One of our objectives is to determine the mass and radius of a selection of stars in correlation to the description of nuclear matter modeled by the EoS. In this way, we can systematically use different EoS parametrizations to determine relevant characteristics of neutron stars. In addition to that, we discuss briefly the use of a Bayesian statistical

model with heteroscedastic errors. This enables the training of statistical models based on EoS generated by different nuclear physics pictures. Due to the various parametrizations present in the microscopic models, the result set of all equation of state has a variance that increases alongside density (heteroscedasticity). This behavior can be captured by models with scattering residuals at different levels of the EoS when trained simultaneously with the NumPyro probabilistic programming library. Here, we use the set of EoS from the Laser Interferometer Gravitational-Wave Observatory (LIGO) as input and handle the data set as a test case.

## 2. The Structure of Neutron Stars

The description of NS comprises both the quantum mechanical and general relativity worlds. The properties of particles that constitute stellar matter are considered via equations of state obtained from quantum mechanics in flat space. The EoS is present in the energy-momentum tensor $T^{\mu\nu}(\rho, P(\rho))$, the bridge to the gravitational/geometric degrees of freedom $G^{\mu\nu}$, through Einstein's general relativity equations

$$G^{\mu\nu} \equiv R^{\mu\nu} - \frac{1}{2} g^{\mu\nu} R = 8\pi T^{\mu\nu}. \tag{1}$$

For a perfect fluid energy-momentum tensor and for a static spherical symmetric spacetime, Einstein's field equations lead to the hydrostatic equilibrium equation, well-known as the Tolman–Oppenheimer–Volkoff equation [37,38]. This equation reads in natural units

$$p' = -(\rho + p) \frac{4\pi p r + m/r^2}{(1 - 2m/r)}, \tag{2}$$

where the prime indicates radial derivative and $m$ is the gravitational mass enclosed within the surface of radius, i.e.,

$$m' = 4\pi\rho r^2. \tag{3}$$

To solve this system, one needs to add to it an EoS ($p(\rho)$) and use the boundary conditions

$$m(r)|_{r=0} = 0, \quad p(r)|_{r=0} = p_c \quad \text{and} \quad \rho(r)|_{r=0} = \rho_c, \tag{4}$$

where $p_c$ and $\rho_c$ are the pressure and density at the center of the star. The numerical integration of Equation (2) follows the pressure decrease as one moves away from the center, and it is stopped when the condition

$$p(r)|_{r=R} = 0 \tag{5}$$

is reached at the surface of the star $R$. The integration of the profile density

$$M(R) \equiv 4\pi \int_0^R r^2 \rho(r) dr \tag{6}$$

provides the total gravitational mass of the star $M$. The resulting M–R relation can be compared to data from astronomical observations. Once the EoS is provided, the global properties of the neutron stars can be obtained. However, until recently, the uncertainties in the mass–radius relationship were significantly large so that almost any EoS could describe the same stellar structure.

The NS can be subdivided into many layers with different theories. Roughly, we can have four regions for the interior: the inner and outer core and the inner and outer crust. For the exterior part, an atmosphere with plasma governed by strong magnetic/electric fields is frequently assumed. The theories to describe the interior span many-body theories of highly-dense strongly-interacting systems, nuclear many-body theories in the high density-temperature regime, atomic structure and plasma physics [39]. We recall that, due to all these different regimes/densities, only the outer crust is well understood, since one can compare with experimental data of atomic nuclei. Around the nuclear saturation

density and above, the constraints become too fragile, allowing for many descriptions of the NS interior: for the outer core, $npe\mu$ (neutron-proton-electron-muon) plasma, and for the inner core, many possibilities such as fermion/boson condensation, hyperons, pion/kaon condensation, strange quarks surrounded by hadronic matter and so on. This complex puzzle calls for an extension of our knowledge about the many-body physics regimes and should lead to models being able to describe a large variety of environments all at once.

## 3. The Equation of State

The description of the outer crust of neutron stars is well accepted to be given in terms of hadronic matter up to the saturation density $\rho_0 \simeq 2.8 \times 10^{14}$ g cm$^{-3}$. This limit reflects the validity of well-established nuclear structure models that were developed to describe properties of heavy atomic nuclei on Earth. When one goes beyond $\rho_0$, more sophisticated degrees of freedom, as mentioned in the previous section, have to be considered. These extra variables make a universal and simultaneous description of systems with such large range density profiles a challenging task. The microscopic constraints are, so far, just a few, and consist of electric neutrality, beta equilibrium and $dp/d\rho$ being always positive and less than the speed of light as well as well defined with $p \geq 0$.

Generally speaking, the different sets of EoS can be separated according to the compressibility (soft and stiff) of the nuclear matter, which is related to the speed of sound. Among the several microscopic methods for EoS generation, we cite perturbation expansions within the Brueckner–Bethe–Goldstone theory, perturbation expansions within the Green's function theory, variational methods and effective energy-density functional and relativistic mean-field (RMF) models [39–44]. Point-coupling and non-relativistic models employing well-known nuclear interaction such as Skyrme and Gogny are also used [45–51]. Two ways are frequently seen in the literature to constrain the EoS models: approaches that consider the physics around $\rho_0$ [52–54], or models that are aimed specifically at systems such as binary neutron star mergers, e.g., using LIGO-VIRGO observational data for the mass–radius of NS to extract the embedded EoS models [35]. In general, the EoS are generated through these models using parameters adjusted to reproduce fundamental physical quantities and are listed in tabulated data, i.e., there are many models and many codes/ways to generate them.

The phenomenological models have the advantage of being easily parametrized and can generate EoS that reproduce the M–R diagrams, offering simpler representations of sophistical microscopic calculations. These are the so-called representations of the EoS, which are basically two: the piecewise polytropic [36,55–60] and spectral representations [61,62]. Here, we focus on models of the first kind.

*Piecewise Polytropic Representation*

The piecewise polytropic model consists of a connected set of polytropic equations, effectively power-law-like functions, with different adiabatic indices to account for the softness/stiffness of the EoS at a given density regime. The indices are free parameters in most of the cases when one considers this kind of parametrization. The density where the transition between the polytropes takes place can also be used as a free parameter, specifically at highly dense regions [63]. The polytropic representation can yield macroscopic observables for a wide range of EoS with only a few parameters. The stellar structure maps the EoS parameters to gravitational mass, radius, moment of inertia and other global properties. This representation has been extensively used in NS studies and gravitational wave simulations [64–67] and can be tested using astronomical data such X-ray, gamma and gravitational waveforms. The representations can also be very useful when dealing with modified gravity such as $f(R)$ [68,69] and other alternative theories where a coupling between geometry and matter could introduce corrections in the energy density and, therefore, requires an analytical representation to model the stellar structure [70].

The piecewise polytropic parametrization of the EoS can be written as [36]

$$p(\rho) = K_i \rho^{\Gamma_i}, \tag{7}$$

where $\Gamma_i$ are the adiabatic indices and $K_i$ are the strength constants.

Here, to model compact stars, we utilize a piecewise polytropic EoS with three basic constraints: (i) Due to the continuity of pressure at the transition points, we impose (with $i > 0$)

$$K_i = K_{i-1} \rho_t^{\Gamma_{i-1} - \Gamma_i}, \tag{8}$$

where $t$ are the EoS transition points. (ii) Causality must be respected, i.e., the velocity of sound is $v_s \leq 1$. (iii) We keep fixed the description of the first part of the EoS (see Figure 1) in the parametrization SLy4 of the nuclear interaction Skyrme-type model.

The transition to the first polytrope with $\Gamma_1$ takes place at the low-density region $\rho_1$, which ensures a good representation of the stars' outer crust. The SLy equation of state describes very well the nuclear matter and matches the BPS and HP94 based on experimental nuclear data, e.g., see Figure 1.3 of Ref. [71]. The transition to the second polytrope with $\Gamma_2$ happens at a higher density, $\rho_2$. One note is that, at the highest densities, much higher than the ones in the cores of neutron stars, the matter goes to unconfined quarks and gluons and the EoS of Quantum Chromodynamics (QCD) becomes accessible to perturbation theory. Therefore, we keep solutions of pressure and density consistent with parametrized QCD (pQCD) [72,73].

The two polytropic combinations of (7) have the set of parameters $\{\Gamma_1, \Gamma_2\}$ and the transitions taking place at $\rho_1$ and $\rho_2$ (see Figure 1). We take the transition points as hyperparameters, while the adiabatic indices are analyzed with statistical methods in relation to data from astronomical observations; the index specifies the stiffness of the EoS in the intervals and restricts the mass–radius relationship. We provide a brief discussion about the challenges involved when $\rho_2$ is reduced to a simple parameter to be adjusted together with the adiabatic index.

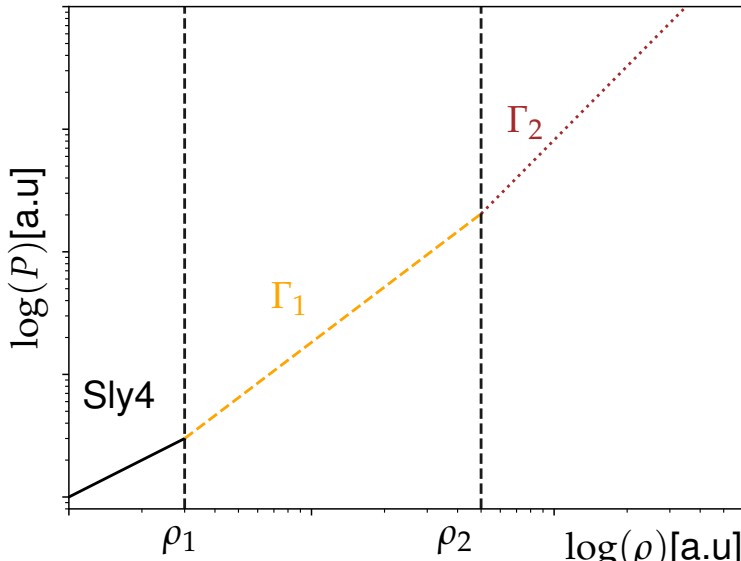

**Figure 1.** Piecewise model representation of the equations of state with the polytropic Equation (7). The black continuous line represents the SLy4 EoS region, the orange dashed line the EoS for first politrope and the red dotted line the EoS for the last politrope. The vertical lines represent the transition points $\rho_1$ and $\rho_2$ of each piece of the EoS.

## 4. Markov Chain Monte Carlo and Bayesian Inference

Markov Chain Monte Carlo (MCMC) is a convenient numerical way to stochastically explore a space of parameter values with high probability and provides good expectation

estimates for model variability. This is basically the point of any Bayesian inference quantification, summarized in terms of mean and variance values. Assuming a distribution $F$ for a given parameter with mean value $q_t$, an associated uncertainty $\sigma_t$ and with a transition probability $K(q'|q)$, we can write

$$F(q) = \int dq' F(q') K(q'|q). \tag{9}$$

Since the form of $F$ is preserved, the algorithm can start from any point $q'$ where the convergence to the typical parameter space region is guaranteed. Sampling from a prior distribution $F$ and employing a simple Metropolis algorithm with $t = 0, \ldots, t = T$ iterations to construct the Markov chain $[q_{t=0}, q_{t=1}, \ldots, q_{t=T}]$, one approximates the posterior distribution given a sufficient large number of steps $T$. The sampled value $q$ is accepted according to the probability

$$\Pi = \frac{f(\chi^2(q_{t+1}))}{f(\chi^2(q_t))}, \tag{10}$$

where $f(\chi^2) = e^{-\chi^2}$ is the likelihood function with $\chi^2 = \left( \left( x_{\text{theory}} - x_{\text{obser}} \right) / \sigma_{\text{obser}} \right)^2$. More details and algorithms can be found in Refs. [74–76]. One of the key ingredients is the definition of the cost (error) $\chi^2$ function

$$\chi^2 = P_R + \chi^2_{1.44 M_\odot} \times \tau_{1.44 M_\odot} + \chi^2_{2.14 M_\odot} \times \tau_{2.14 M_\odot}, \tag{11}$$

where

$$P_R = e^{(-(r_{\max} - R_i) 10)} + e^{((r_{\max} - R_f) 10)} \tag{12}$$

with $r_{\max}$ being the radius that corresponds to the star with maximum mass. Equation (12) is a penalization function to ensure that, for the maximum mass generated by the EoS, the star's radius is limited between $R_i = 8$ km (roughly Buchdahl limit) and $R_f = 13$ km (we expect that, for massive starts, it should be at least the same as a star of 1.44 solar mass for a very stiff EoS). Both $\chi^2_{1.44 M_\odot}$ and $\chi^2_{2 M_\odot}$ are associated to the data points we include in our analysis. At the mass of $1.44 M_\odot$, we also include the radius information of the J0030+0451 pulsar with $R_{1.44 M_\odot} = 13.02$ km with $\sigma_{1.44 M_\odot} = 1.1$ km. The last term of Equation (11) defines the upper limit of our target phase space limited to only the mass of PSR J740+6620, $M = 2.14 M_\odot$ and $\sigma_{2.14 M_\odot} = 0.1$. We have also included $\tau_{1.44 M_\odot} = 10^2$ and $\tau_{2.14 M_\odot} = 1$ to control target weights separately (these numbers were kept fixed in this paper, otherwise mentioned). We found this choice of parameters to be optimal for good convergence and sensibility of the algorithm to small changes in the MCMC steps.

In this work, the algorithm will attempt to minimize $\chi^2$ regarding a defined set of observational data from *NICER*, LIGO-VIRGO and massive pulsars and access the variability of piecewise polytropic models for the mid- and high-density stellar regions. The set of parameters adjusted are assumed to be uncorrelated with uniform prior distributions but limited within [1,10] for $\Gamma$'s values (the upper range limit is large enough to prevent bound limitations, as seen in our calculations) and $[\rho_0, 5\rho_0]$ when $\rho_2$ is included in the MCMC. For the low-density region, below the nuclear saturation density, which is well understood in terms of hadronic matter, we use the SLy4 equation of state and glue it to the first polytrope at $\rho_1$. For $\rho > \rho_1$, the intermediate–high-density portion of the EoS is modeled with adiabatic index $\Gamma_1$. This segment of the EoS is not fully understood with our current knowledge of microphysics, representing a density region where EoS variability can be studied. Finally, the transition to the second polytrope with adiabatic index $\Gamma_2$ (right side of Figure 1) happens at $\rho_2$ and represents the densest part of the EoS. The value defining this transition is difficult to be estimated since the physics of highly dense interacting matter is yet not known in detail, and, therefore, it can be arbitrarily chosen.

As a first step, we start by looking at the MCMC convergence of our calculations. To exemplify it, we take the case of $\rho_1 = \rho_0$ and $\rho_2 = 2\rho_0$. Figure 2 shows the results of our algorithm for $T = 5 \times 10^4$ iterations obtained with two different initial values for $\Gamma_{1,2}$. Figure 2a,d show the trace of the chain during the sampling process, and the respective moving average computed with length = 1000 is also shown. The posterior distributions are given in Figure 2b,e. Their mean values are $\langle \Gamma_1 \rangle = 3.6$ and $\langle \Gamma_2 \rangle = 2.3$. Here, the posterior of $\Gamma_2$ has a broader shape when compared to $\Gamma_1$. This behavior was verified for all initial conditions tested and will be discussed later on in this paper. The autocorrelation function is shown to vanish quickly as one compares values, with a lag of about 80 iterations in the chain.

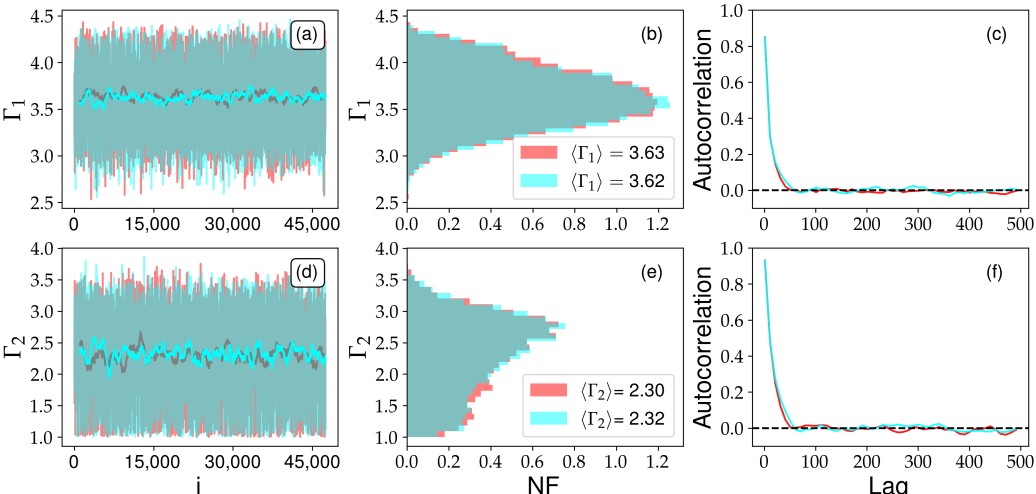

**Figure 2.** MCMC chain (**a**,**d**) and respective posterior distributions (**b**,**e**) for both $\Gamma_1$ (upper panels) and $\Gamma_2$ (lower panels) obtained with $\rho_1 = \rho_0$ and $\rho_2 = 2\rho_0$ with $T = 5 \times 10^4$ iterations. The deviation of the posterior average values along the chain is small $\sigma_{P(\Gamma)}/T \approx 0.02$, indicating that a small portion of the posterior distribution is due to sampling error. This can be visualized in the moving average of the MCMC chains. Autocorrelation functions are shown in (**c**,**f**).

We show in Figure 3 the Gelmen–Rubin (GR) diagnostics [77] as a test for the statistical convergence of our calculations. Here, one expects that if the chains obtained with different initial conditions have converged, they should necessarily be similar to one another. Their similarity is given by the GRc coefficient,

$$\text{GRc} = \sqrt{\frac{\mathcal{S}(\Gamma)}{\bar{\sigma}^2}}, \tag{13}$$

where $\bar{\sigma}^2 = \sum_k^M \sigma_k^2/T$ is the within-*M*-chain variance (*M* chains obtained with different initial conditions) and

$$\mathcal{S} = \left[1 - \frac{1}{T}\right]\bar{\sigma}^2 + \frac{1}{T}\sigma_0^2 \tag{14}$$

is the weighted averaged variance. The between-chain variance is given by

$$\sigma_0^2 = \frac{T}{1 - M}\sum_k^M \left[\langle \Gamma \rangle - \langle\langle \Gamma \rangle\rangle\right], \tag{15}$$

where $\langle\langle \Gamma \rangle\rangle$ is the mean of all individual chain means. One can easily notice that if Equation (13) provides results close to unity, the variance among the series is small, indicating that the chains are embedded in a stationary posterior distribution.

We have averaged over five runs with different and randomly assigned initial conditions for the parameters $\Gamma_{1,2}$. Figure 3 shows a good numerical convergence with 50 thousand iterations, proving GRc coefficients $\approx 1$ with a burn in of 1000 and a length of 500 are employed. This length is enough to vanish with the correlations within a chain, as can be seen in Figure 2. Based on Figure 3, we set a burn in, i.e., we disregard the first 1000 iterations for every simulation presented in this work. This should remove the transient behavior of the posterior distribution and leave us with 49 thousand EoS to represent each model.

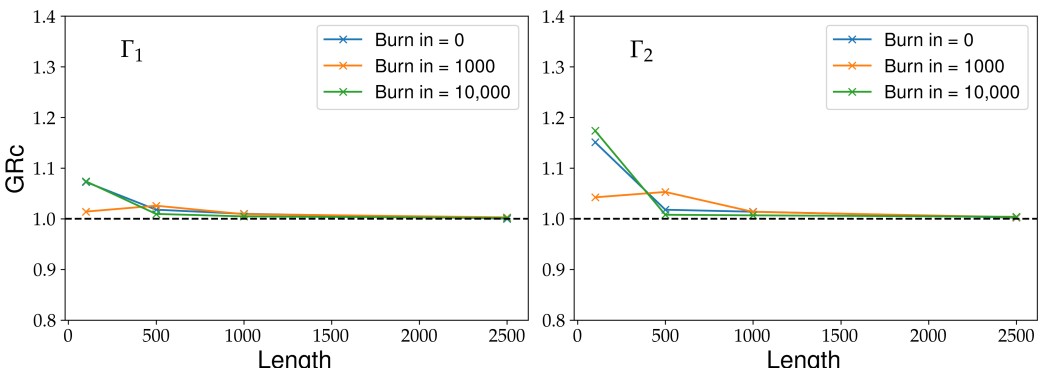

**Figure 3.** Gelmen–Rubin diagnostics for the case shown in Figure 2 for both $\Gamma_1$ (**left**) and $\Gamma_2$ (**right**). The Gelmen–Rubin coefficient $\approx 1$, the black dashed line, shows the numerical convergence of the MCMC algorithm.

Besides the first model (MD1 model), which we have considered for the MCMC convergence test, we also have considered two additional combinations of $\rho_{1,2}$ as hyperparameters; we summarize these values in Table 1. Once the transitions points $\rho_{1,2}$ are defined, we can systematically find $\Gamma_{1,2}$s using the MCMC technique and calculate the constant $K$ for each individual polytrope using Equation (8), which will give us the full the description of the EoS. The use of the MCMC algorithm is based on solving the T.O.V. equation and matching the output with the observational data: mass–radius data from *NICER*, the pulsar J0030+0451, with an error of $\sigma_{M\ [M_\circ]} \approx \langle 0.15 \rangle$ and $\sigma_{R\ [km]} \approx \langle 1.1 \rangle$; the massive pulsar PSR J0740+6620 with an error mass of $\sigma_{M\ [M_\circ]} \approx \langle 0.1 \rangle$, not having an observational radius for the massive pulsars, i.e., a low narrow constraint for M–R, leads to a large range of possibility for EoS that explain stars above two solar masses. There are also less narrow constraints, such as the GR limit.

In total, we have considered three combinations of hyperparameters schemes. Our MD#s provide a large variety of equations of state, i.e., different densities and pressure profiles with the respective stellar global properties. The models are summarized in Table 1, where the color scheme used in the figures is also provided.

**Table 1.** Summary of the hyperparameters and averaged adjusted polytrope indices for piecewise EoS models. We employed 49 thousand EoS for each case. $\rho_0 = 2.8 \times 10^{14}$ g cm$^{-3}$; $n_0 = 0.17$ fm$^{-3}$ is the nuclear saturation density.

| Label | $\rho_1$ | $\rho_2$ | $\langle \Gamma_1 \rangle$ | $\langle \Gamma_2 \rangle$ | Color |
|---|---|---|---|---|---|
| MD1 | $\rho_0$ | $2\rho_0$ | 3.6 | 2.3 | purple |
| MD2 | $\rho_0$ | $3\rho_0$ | 3.2 | 2.2 | blue |
| MD3 | $\rho_0$ | $5\rho_0$ | 3.1 | 4.7 | green |

In Figure 4, we present in detail the model used to test the convergence of our calculations, the model MD1 from Table 1. On the left side of the figure we have the mass–radius relation corresponding to the EoS of this first model. In this sub-figure, we also have the LIGO-VIRGO mass–radius region, in red and orange, constrained by the gravitational wave event GW170817 [6,78]; this constraint was the first one to have a radius associated with the mass tightly constrained, since the previous observations for neutron stars were using electromagnetic bandwidth, which makes it very difficult to estimate the radius of the NS. After this gravitational wave detection, the radius of a pure nuclear hadronic matter with mass of 1.4 $M_\odot$ was estimated to be $\overline{R}_{1.4}$ = 12.39 km [79]. Afterwards, this constraint was shifted due to measurements of *NICER* for the pulsar PSR J0030+0451 to two values: $M \approx 1.44\ M_\odot$ with an equatorial radius of $R_{\rm eq}$ = 13.02 km [20], and $M \approx 1.24\ M_\odot$ and $R_{\rm req}$ = 12.71 km [21]. This information is highlighted by the dark dots with error bars in the mass–radius diagram. We also present upper limits of mass: a lower mass compact object with $2.50 - 2.67\ M_\odot$ in a binary system detected by LIGO-VIRGO [80]; this unknown object, if taken as an NS, will be a breakthrough, since no nuclear theory for ordinary matter can explain the necessary EoS to generate such a mass in general relativity. Finally, we consider observations of massive pulsars: the extremely massive millisecond pulsar PSR J0740+6620 with a mass of $2.14^{+0.20}_{-0.18}\ M_\odot$ [81] or $2.08^{+0.07}_{-0.07}\ M_\odot$ [82]; and the PSR J2215+5135 with mass $\approx 2.27\ M\odot$ [83]. The measurement of the mass of the source is not so precise, and, if this number is confirmed, the star would be the most massive neutron star ever detected; the two most well-known NS sources are J0348+0432 and J1614-2230 [84,85] with $M = 2.0\ M_\odot$. The mass–radii in purple were generated using the posterior distribution of the parameters obtained by the MCMC constrained by these observations. The curves are expected to cross the region of the *NICER* observation, i.e., a mass of 1.4 $M_\odot$ with the respective radius at the same time reaches massive pulsars as a mass as high as $M = 2.0\ M_\odot$. In the upper left corner, we have the EoS from this model, where we can see a cut-off before reaching the limit from pQCD. In the middle on the right, we have the EoS speed of sound, and, in the lower right panel, the mass vs. central densities of the corresponding mass–radius figure. The dark line represents the SLy4 EoS in all figures and is used for comparison reasons since this EoS can reach massive pulsar, and it is near the *NICER*-LIGO-VIRGO observation. Analyzing the two transition regions, we can see that the EoS is more stiff in the first polytrope. The sound speed increases quickly in the beginning and decreases in the second polytrope (a result of $\Gamma_1 > \Gamma_2$ on average). Comparing MD1 with the SLy4 in the $P(\rho)$ diagram, we see that the large majority of the MD1 EoS lies above Sly4 in the first polytrope region, and well below in the second part. We also noticed that the first polytrope can generate masses near 2.0 $M_\odot$ due to the stiffness of the EoS in this region.

In the two panels in Figure 5, we have the two models MD2 and MD3 from Table 1, respectively. In these two panels, we are considering the transition for the second polytrope at higher densities. These models start to limit the maximum mass reached for the stars, since the second adiabatic index becomes less important, i.e., the stars of interest here can be described within the first region with one polytrope $\Gamma_1$. One should mention that we cannot construct the EoS for higher densities going too close the pQCD limit; if we look at the EoS panel on the upper right side, we see a decreasing cut-off as the transition regions increase. One can also observe a stiff discontinuity in the sound speed. For the first polytrope, the sound speed is well above the conformal limit from QCD $v_s^2 = 1/3$ due to large $\Gamma_1$ values. We remark that, according to massive pulsars [81,82,85] and some theoretical work [8,86–89], the nonmonotonic and sub-luminal $v_s^2 \lesssim 1$ and $v_s^2 > 1/3$ for $> \rho_0$ is most likely the EoS.

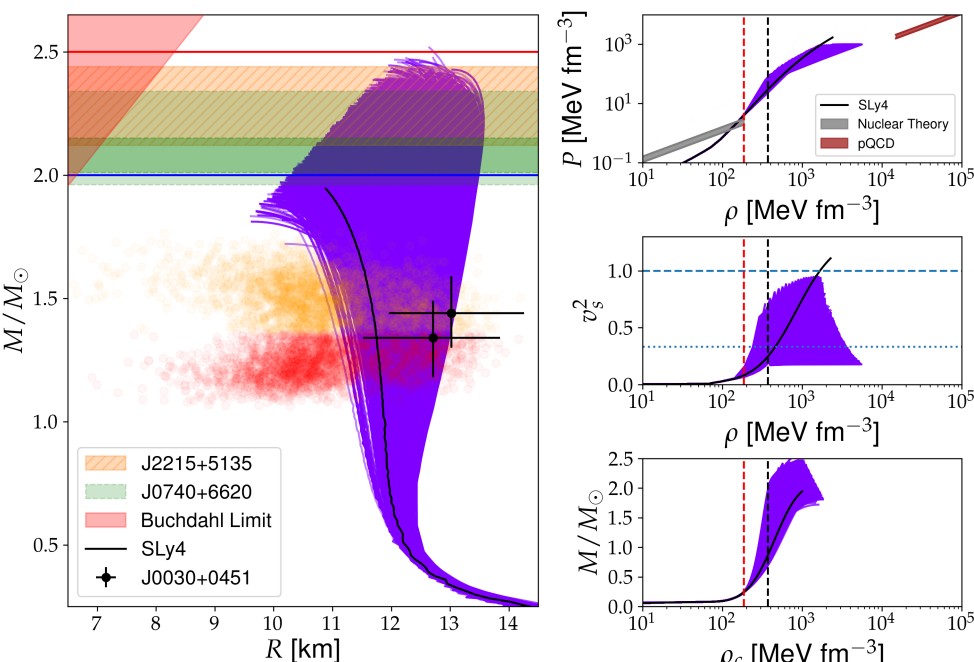

**Figure 4.** On the left side: Mass–radius relationship for the MD1 parametrization from Table 1. The blue continuous line at 2.0 $M_\odot$ corresponds to the two massive pulsars J0348+0432 and J1614-2230. The filled green region represents the pulsar J0740+6620 and the filled dashed salmon region is the pulsar J2215+5135. The red line is the low mass compact object in the binary system GW190414. The dark dots with errors bars are the *NICER* estimations of PSR J0030+0451. The purple curves in the left panel are the mass–radius relationships for the EoS generated by the MCMC algorithm. In the upper right corner, in purple, we have the MD1 EoS generated by the algorithm. In the middle right panel, we have the sound speed, and, in the lower panel, the masses for different central densities. The two vertical lines represent the transition regions, and the dashed-dotted horizontal lines in the middle right panel are the luminal and conformal velocities. Dark lines represent the SLy4 EoS.

We have also performed a test case with $\rho_2$ as a statistical parameter in addition to the $\Gamma$s. This case is shown Figure 6. One can observe that, in general, $\Gamma_1$ is $\gtrapprox 4$, while $\Gamma_2$ is $\lesssim 3$, in accordance with the previous remark. The abrupt decrease in the speed of sound in the second polytrope could be interpreted as a first-order phase transition, i.e., one core of quark-matter [90] for massive stars with $M > 2M_\odot$, since we observe that the sound speed goes to 1/3.

We noticed the difficulty of adjusting $\rho_2$ with these limited observations. For this case, one needs more data regarding the merge of NS–NS to have a better description of the transition regions. Since tidal deformability has been, up to now, associated with the GW event where the value for one NS is still uncertain (it could fall to a black hole), a prior distribution for $\rho_2$ considering the tidal deformability could be included in our model when more observational data becomes available. Another approach would be to have the radius of one of those massive pulsars to constrain the MCMC further in the massive region. For now, with the set of observations that we included here, we noticed that a combination of two polytropic equations and the SLy4 EoS provides good interpretation of the observational data. The number of degrees of freedom around the nuclear saturation density is well established, and, for $\rho \approx \rho_{\mathrm{pQCD}}$, one has to respect the pressure/density of pertubative QCD limits [73,90], not included in other works [91].

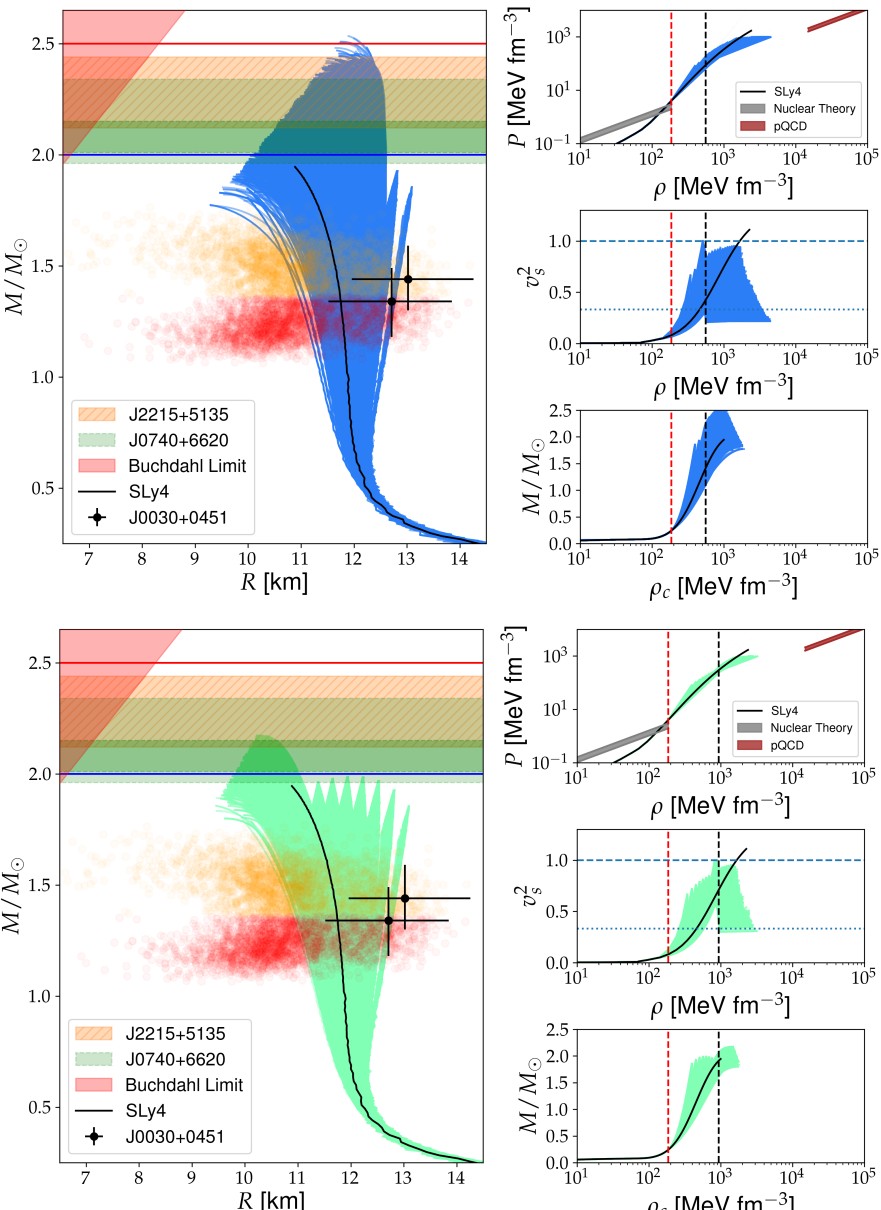

**Figure 5.** Same as Figure 4: in the upper panel, the blue one, the transition regions are $\rho_1 = \rho_0$ and $\rho_2 = 3\rho_0$, and, in the lower, the green one, $\rho_1 = \rho_0$ and $\rho_2 = 5\rho_0$.

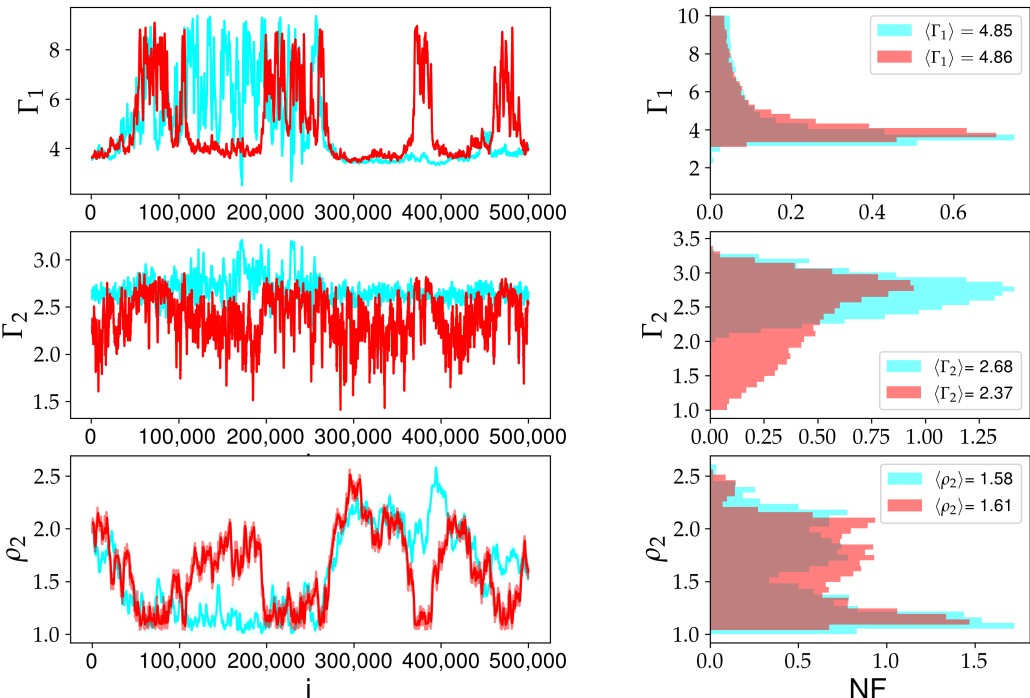

**Figure 6.** Analysis considering $\rho_2$ as a free parameter together with $\Gamma_1$ and $\Gamma_2$. Cyan and red curves show the first transition at $\rho_1 = 0.5\rho_0$ and $\rho_1 = 2\rho_0$, respectively. The intermittent behavior of the series represents "unstable" solutions of the minimization problem, more observational data points are needed here to reduce the variability of the parameters.

## 5. Bayesian Power Regression Model with Heteroscedastic Errors

In this section, we briefly investigate the potential use of a Bayesian Power Regression model with heteroscedastic errors (BPR-HE) to capture the relationship between the density and pressure. The idea here is to train a model that incorporates the associated variance of a large variety of physics parametrizations. This approach could then be constrained by observational data automatically in a physics-informed machine learning strategy. As a preliminary step towards that, we focus on the BPR-HE approach.

Power regression is a non-linear regression model that takes the form $y = ax^b$, where $y$ is the response variable, $x$ is the prediction variable and $a$ and $b$ are the coefficients that describe the relationship between $x$ and $y$. The model can be made linear by simply applying a log transformation: $\log(y) = \log(a) + b \log(x)$. Therefore, one can infer the parameters of a non-linear power regression model via a linear model. With that, our corresponding BPR-HE model is defined as

$$\begin{aligned}
\log(p^i) &\sim \text{Normal}(\alpha \cdot \log(\rho^i) + \beta, s_m \cdot \log(\rho^i) + s_b), \qquad \forall i = 1, ..., N. \\
\alpha &\sim \text{Normal}(\gamma_1, \gamma_2) \\
\beta &\sim \text{Normal}(0, \gamma_3) \\
s_m &\sim \text{HalfCauchy}(\gamma_4) \\
s_b &\sim \text{HalfCauchy}(\gamma_5)
\end{aligned}$$

(16)

where $\gamma_*$ are a set of hyperparameters that are specified by the user. In our experiment, we set all $\gamma$ to 1. N = 65 is the total number of equations of state taken from the LIGO *Lalsuite* [92] library and used as the data set. Essential to the model is the dependence of the standard deviation of the residual to the density variable $\rho$. This is necessary, as the ensemble of EoS from LIGO have an increasing pressure variance regarding densities. Residuals with varying variance are known as heteroscedastic. Figure 7 shows an illustrative example

of heteroscedastic errors and homoscedastic errors, which are assumed in classical linear regression models.

We used the Numpyro probabilistic programming language [93] to implement the model (16). We inferred the values of the unknown parameters in our model ($\alpha$, $\beta$, $s_m$ and $s_b$) by running MCMC using the No-U-Turn Sampler (NUTS) [94] for 10,000 warm-up samples and then collected 1000 posterior samples to represent our model parameter's posterior distribution.

Figure 8 shows posterior samples of the BPR-HE model (in yellow) and the EoS from LIGO. Notice that the model captures the increasing variance of the pressure as the density increases. This is due to the heteroscedastic errors in the model. The BPR-HE provides smooth functions, rather than the piecewise polytropic approach described in the previous section. The abrupt transition points of the MD# models can reduce the accuracy of the description of local speed of sound, a problem already discussed in [36].

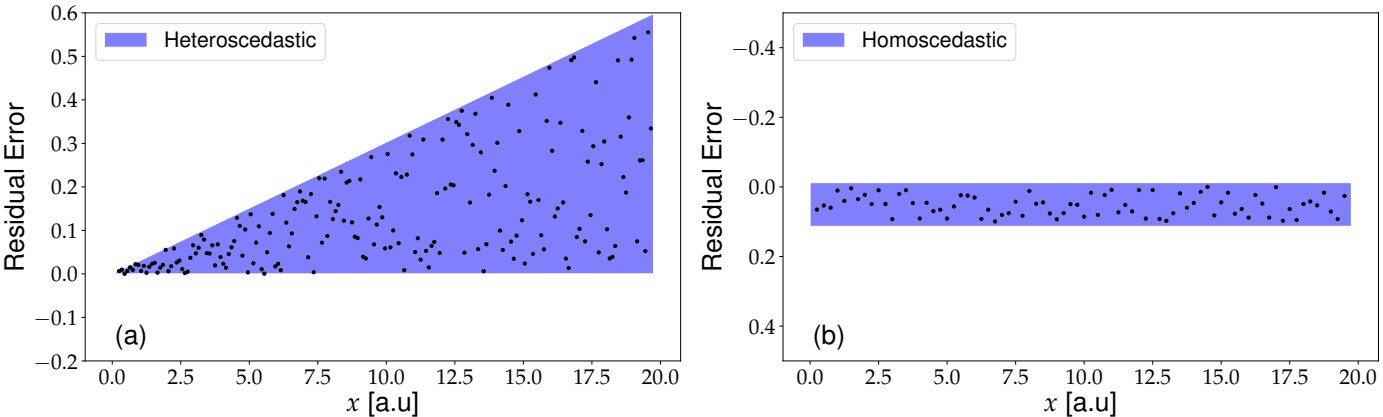

**Figure 7.** Representative example of heteroscedastic (**a**) and homoscedastic (**b**) residuals. Notice how the variance of the residuals changes with the value of $x$ for the first, while it remains constant for the second.

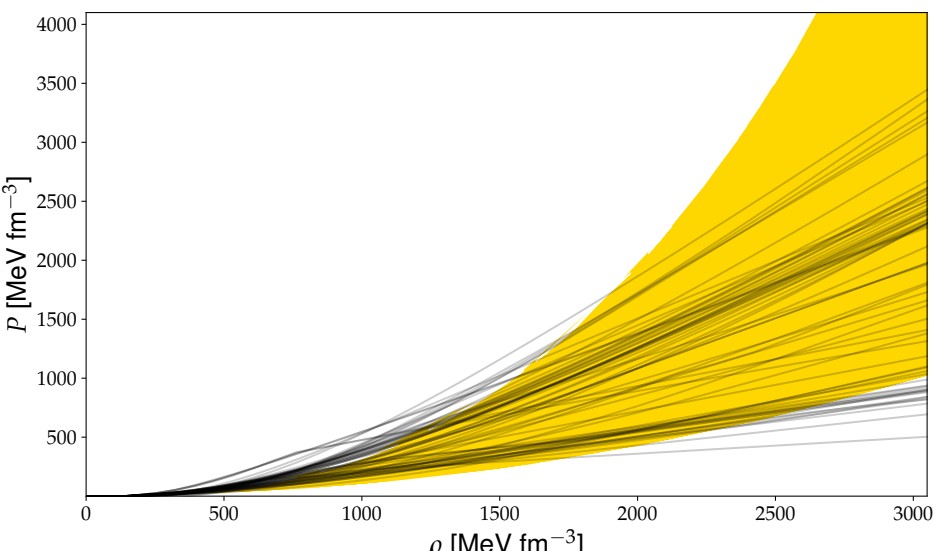

**Figure 8.** Bayesian Power Regression model heteroscedastic errors. Black solid lines are the 65 EoS from the LIGO *Lalsuite* [92] data set, while the yellow ones are posterior samples generated by the BPR-HE model.

From Figure 8, we observe that the BPR-HE model is a promising approach to be used to model the relation of density and pressure. This information can be used to estimate

uncertainties of the maximum radius and mass of the neutron stars via T.O.V. equations. We left the use of BPR-HE-generated EoS in the T.O.V. equation for future works and have restricted this work to a feasibility analysis of such approach.

## 6. Conclusions and Perspectives

In this work, we have performed a Bayesian exploration of parameters of piecewise polytropic equations that can represent the equation of state of neutron stars. We have considered a set of two-piece polytropic models glued to the Skyrme-type EoS known as SLy4 that represents well the densities below the nuclear saturation density. The two polytropic were used to model the high-density region, recent observational data from massive pulsars as well as the results from LIGO-VIRGO/*NICER*. Polytropic piecewise models with few parameters are known to be good representations for modern theoretical EoS and can reproduce global features of neutron stars such as mass, radius, moment of inertia and so on (e.g., see Refs. [29,35,36,59,95,96]). Commonly used in the literature, this phenomenological approach is applied in a broad research context, from numerical solutions of rotating relativistic stars/merger simulations [67,97,98] to modified gravity [68,70,99,100] studies. Recently, they have been successfully applied to constrain the dense matter equation of state of neutron stars supported by observations [35,101–104].

We considered three models, i.e., three combinations of hyperparameters, which led to a variety of combinations of adiabatic parameters being globally adjusted. The performed analysis has accounted for different astronomical observational sources, in a joint constraint, and accessed the variability of our models. The constrained representation of the EoS was compared with observations of neutron stars. The parameters studied within the MCMC were open to allow soft and stiff EoS, and different transition point regions in compact stars were suggested. From massive pulsar observations, one knows that the EoS should be stiff enough to support stars above two solar masses, and it seems to explain the *NICER* observation as well as [22], but soft EoS also favor the GW170817 event [12] while permitting a quark core in the NS core [90]. In this case, near the core of the star, one would expect a phase transition and notice the conformal limit, i.e., have a sound speed that goes to $v_s = 1/3$. Changing the transition point definitions allow us to investigate the sensibility of such a change in the MCMC and see the behavior of the adiabatic indices in the possible EoS generated.

As we have observed from our results, the transition between the two polytropes at higher densities directly affects the values of the two-adiabatic index, and, therefore, the speed of sound. For low-density transitions, the speed of sound goes to higher values monotonically, indicating that we have a hadronic matter with $v_s > 1/3$ that could be strongly interacting and nonconformal. If we set the transition to a higher density, we observe a softening of the EoS after the transition, and, in this case, the system can produce a nonmonotonic behavior for the speed of sound. Maximum values can take place along the first polytrope region that can be higher than the conformal limit $1/3$. The sharp change towards the second region of the EoS could then indicate a first-order transition, opening the possibility of a hybrid star with a quark core. However, we still need more study about that, since the second adibatic index starts to lose importance for these cases since most of the two solar mass stars are reached within just the first polytrope.

From our adopted schemes, the two models, MD1 and MD2, can represent very well stars for mass around 1.4 $M_\odot$ and radius of $\approx 12$ km, i.e., the mass–radius observational region of LIGO-VIRGO binary NS merger and the PSR J0030+0451 constrained by the *NICER* experiment. These models can also explain massive pulsars with mass above 2.0 $M_\odot$ as the two pulsars J0348+0432 and J1614-2230. One of the models, MD1, can even be very close to explaining an unknown object with a mass of 2.5 $M_\odot$ in a binary system, detected by LIGO-VIRGO. As one can notice, the model MD1 yields almost the same radius for different masses, almost a limit for the adiabatic index.

We believe this study of piecewise equations with recent observations can help to define different matter regimes for highly dense matter. In previous works [105,106], we

analyzed correlations in the microphysics of many EoS and in the global properties. We have studied two separated spaces, and now we can bring these two complementary studies together in a full picture and, employing statistical and machine learning tools, shed light on the path to understand the EoS of neutron stars. We attempted to use a reduced number of parameters compared to other works [36] and also employ more astronomical/theoretical constraints.

Before closing, we would like to comment on some challenges and remarks on modeling EoS coming from many nuclear models with different parametrizations. We understand that more data with respective uncertainties are required to reduce the parameter space and variability of the models. Upper and bottom limits for both mass and radius have to be included, since a single star can be explained with different EoS, but the limits should always be respected. We expect that, when more data becomes available, the degrees of freedom will be reduced and some star interior specificity can be studied in more detail. Statistical models, such as regression models with heteroscedastic errors, for example, have the potential to best represent a set of different physics included in a variety of equations of state. The Bayesian Power Regression model with heteroscedastic errors (BPR-HE) is a flexible model, but we faced difficulties with it due to the nature of the data. In the model, the variance of the errors varies linearly with the density value, which might not be appropriate, as the $s_m$ parameter has shown to be very sensitive and difficult to infer. We had to resort to a forceful (informative) before to stabilize the inference. Another point is that using a single power regression model to describe all EoS might be too restrictive given the diversity of physical models. We believe that a mixture regression model, composed of several power regressors, will bring more flexibility. These points will be the focus of future research steps, as well as the tension brought by the Lead Radius EXperiment (PREX-2) results with astronomical data.

**Author Contributions:** Conceptualization, E.V.C., R.V.L., A.R.G. and C.A.B.; methodology, E.V.C., R.V.L., A.R.G. and C.A.B.; software, E.V.C., R.V.L., A.R.G. and C.A.B.; validation, E.V.C., R.V.L., A.R.G. and C.A.B.; formal analysis, E.V.C., R.V.L., A.R.G. and C.A.B.; investigation, E.V.C., R.V.L., A.R.G. and C.A.B.; resources, E.V.C., R.V.L., A.R.G. and C.A.B.; data curation, E.V.C., R.V.L., A.R.G. and C.A.B.; writing—original draft preparation, E.V.C., R.V.L., A.R.G. and C.A.B.; writing—review and editing, E.V.C., R.V.L., A.R.G. and C.A.B.; visualization, E.V.C., R.V.L., A.R.G. and C.A.B.; supervision, E.V.C., R.V.L., A.R.G. and C.A.B.; project administration, E.V.C., R.V.L., A.R.G. and C.A.B.; funding acquisition, E.V.C., R.V.L., A.R.G. and C.A.B. All authors have read and agreed to the published version of the manuscript.

**Funding:** This research was partly funded by the U.S. Department of Energy (DOE) under grant DE-FG02-08ER41533 and by the LANL Collaborative Research Program by Texas A&M System National Laboratory Office and Los Alamos National Laboratory. In addition, this research was partly funded by UNIANDES University. The work at Brookhaven National Laboratory was sponsored by the Office of Nuclear Physics, Office of Science of the U.S. Department of Energy under Contract No. DE-AC02-98CH10886 with Brookhaven Science Associates, LLC.

**Data Availability Statement:** This is a theoretical work and the data used are public in the cited and respective references.

**Acknowledgments:** We acknowledge the organizers of the conference "The Modern Physics of Compact Stars and Relativistic Gravity 2021" and Armen Sedrakian, the editor of the special issue "Selected Papers from "The Modern Physics of Compact Stars and Relativistic Gravity 2021"" for the contribution invite.

**Conflicts of Interest:** The authors declare no conflict of interest.

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
