# Peer review of "Bayesian Exploration of Phenomenological EoS of Neutron/Hybrid Stars with Recent Observations"

_2571-712X, doi:10.3390/particles6010011_

Round 1
Reviewer 1 Report
# Bayesian Inference of Phenomenological EoS of Neutron Stars with Recent Observations
### Emanuel Chimanski, Ronaldo Lobato, Andre Goncalves, Carlos Bertulani
Phenomenological representations of the equation of state (EoS)
of neutron stars (NS) represent a useful tool in current day nuclear astrophysics. Such representations are able to faithfully
reproduce complex EoSs with a limited number of parameters,
and -- if informed by observations -- can be employed to constrain
the pressure-density EoS plane, and in turn shed light on the
nature of dense matter.
In this paper, Chimanski and collaborators employ the piecewise-polytropic (PWP) representation introduced in Read et. al. (2009) to construct five distinct sets (or ``models") of EoSs. Each set is then mapped into mass-radius curves by solving the stellar structure TOV equations, and a posteriori compared to observational data.
Unfortunately, I found the paper lacking in several aspects.
The manuscript suffers from poor writing and unclear presentation of methods, scope and results. Many sentences are filled with typos and imprecisions.
I noted below just some of the typos I found, and urge the authors to significantly improve and rework **all** sections of the paper.
Further, in my opinion the work does not represent a particularly useful addition to the literature on the topic. Due to the very limited parameter space that has been explored, the proposed model(s) can not be used as priors
to drive future analyses (which would be the main application for such a study).
Finally, I also fail in finding novelty in this work. Previous publications (see e.g. Sec.II of 2012.12151) have performed analyses of a similar kind, but with a larger, more generic set of EoS, taking also into account observational constraints on the EoS at the time of sampling.
In summary, I can't recommend this paper for publication in its current form. Should the authors improve the the quality of the presentation and extend the analysis to a wider parameter space (in order to properly study correlations between parameters), I'd be happy to reconsider this statement.
### Additional Comments
* There is a typo in the title of the manuscript, "phenomenologycal"-->"phenomenological"
* There are a few typos in the abstract: "three-piecewised","equation of states", "inputs"
* The acronym "EoS" is used but not defined
* vital sources --> remove vital
* they have been discovery --> their existence has been known for ... or similar
* its internal --> their internal
* The sentence starting with ``part of the challenge'' is unclear and should be rewritten
* ``Those constraints' --> the authors do not discuss any constraint in the previous sentence(s)
* All this information, are --> All this information *is* (also, there is no need for a comma).
* the microphysics can to be constrained as well --> the microphysics can be constrained as well
* "The GW170817 event, for example, besides the breakthrough of being the first gravitational wave detection" --> GW170817 was NOT the first GW detection. It was the first detection **of a BNS system** via GWs. Correct this sentence.
* "The association with electromagnetic counterparts of the event, lead to the first time to a joint-constraint" --> "The association of the event with an EM counterpart led to the first joint GW-EM constraints on the NS EoS"
* "Using the tidal deformability parameter" --> Tidal parameters are not defined or introduced. When doing so, one should also cite at least 0709.1915 and 0906.0096
* "The increment in observational data, has helped" -->no comma
* What are "intermediate densities"? Quantify
* made possible with modern computing resources. --> made possible by
* "We adjust the position of each piece of the EoS" --> What does it mean to adjust the "position?" Please rewrite and clarify.
* "...are considered via equation of state obtained..." --> are considered via equations of state obtained...
* When discussing piecewise polytropes, the authors should also cite 2008.03342
* What was the rationale behind the choices of values for \rho_{1->2} and \Gamma_{1,2}
shown in Tab. 1? And why was sigma chosen to be 0.01?
* Following up on the previous point, the algorithm employed by the authors is a key part of this paper, and should thus be discussed accordingly. In detail, the authors should clearly motivate their choices of priors, likelihood and acceptance function.
* Am I correct in understanding that the MCMC sampling performed by the authors does not enforce observational constraints on the EoS at the time of sampling? If so, the title of this paper appears to be quite misleading to me, as no observational data was actually used during the bayesian inference.
* The authors state that "In total, we have 5 combinations of parameters schemes. Our MD#s, provide a large variety of equation of states". I disagree with their assessment.
Looking at Fig.3 and 4 there are clearly large regions of the p-rho and M-R parameter space that the models chosen by the authors do not cover.
The constraints imposed on the adiabatic indices Gamma_{1,2} are so tight that if the authors had used only the median EoS listed in Tab. I, rather than the full sets that they built, the results of this paper would not have been significantly different. The correlations between the parameters of the phenomenological representation are neglected because of this: I believe that there are regions in the (Gamma_1, Gamma_2) space that could fit the observed data, but are not being taken into account in this study. Similarly, there could be regions in the (Gamma_1, Gamma_2, rho_{1->2}) space that could explain the data, but they are similarly being neglected due to the decision to fix the transition density.
* As mentioned previously, the authors should compare and contrast their study to Sec. II of 2012.12151. To my understanding, the authors' work is very similar to the paper of Godzieba and others, but more limited in parameter space and number of equations of state explored.
* In the conclusions, the authors state that "the parameters found here for the piecewise equations that represent modern EoS, will be used to constraint nuclear models, i.e., the parameters of many-body model." Apart from the typo (constraint-->constrain), as said before I believe that the models used by the authors do not cover a large enough parameter space to meaningfully represent the NS EoS compatible with observations. Therefore, I do not think that they can or should be
used to impose further constraints on nuclear models.
Author Response
We have reviewed the entire manuscript and addressed all issues pointed (all the grammar issues were addressed), in particular: the method's presentation, the scope of the work, and the results. We highlighted the contribution of our study to the literature as we considered new observations and constraints, and provided comparisons with previous works. In summary, we made the manuscript read more clearly as well as the addition to improve our models with more constraints. We have also adjusted the title.

Reviewer 2 Report
Bayesian statistics is a very powerful tool to constrain the equation of state (EoS) of neutron stars (NSs) by combining different kinds of astrophysical observations. In the recent past many authors have successfully used it by constructing different types of parametric and nonparametric descriptions of NS EoS and this led a better understanding on NS properties to the community. In this work, the authors claim to use Bayesian statistics to infer the EoS of NS with the recent observations. Unfortunately I find some serious flaws in their analysis and therefore, I cannot recommend this article for publication. My reasons are as follows:
- The authors use 3-piece polytropic representation of NS EoS in which they always fix the polytropic index used in lower densities. This representation of NS EoS is different from the originally proposed piecewise-polytope parameterization (PHYSICAL REVIEW D 79, 124032 (2009)). In their analysis Read et al. had shown that at least 4 free parameters are needed to accurately fit a wide-variety of theoretically modelled EoSs of NSs. Therefore it is not justified to use a smaller number of parameters without showing how accurately they can fit the theoretically modelled EoSs of NSs. Otherwise, the result is erroneous.
- Based on their parameterization, authors only choose five different combinations of NS EoS models using different combinations of two polytropic indices (Gamma1 and Gamma2) and the transition density between them. Then they assume if the observation data provide an error uncertainty of 0.01 on Gamma1 and Gamma2, how well we can recover them for those five cases. The results are shown in Figure 2. But it does not make sense how the authors can assume an error uncertainty of Gamma1 and Gamma2 as observations! Observations provide us measurement of NS mass, radii, tidal deformability etc. Using those quantities as observables, one should put constraints on the respective parameters of EoS models. Also instead of choosing only five EoS models, authors must use the whole parameterization and compute joint posterior of them using hierarchical Bayesian statistics.
- I have doubts about the convergence of their MCMC algorithm. They claim N=10000 iterations are enough to assure the convergence of results. But this is not the way to check convergence of MCMC chains. I would rather suggest the authors to use Gelman-Rubin convergence diagnostic or some other similar algorithm. Maybe it would be easier to use a package like EMCEE which has an in-built convergence testing algorithm.
To summarise in order to constrain the EoS of NSs, one should take mass, radii, tidal deformability etc as observables. Then considering a physics-motivated priors on EoS parameters, we should look into their joint posterior after including the observation data using a hierarchical Bayesian statistics. I would recommend the authors to look into some standard papers carefully on this subject:
a. G. Raaijmakers et al., Astrophys. J. Lett. 893, L21 (2020)
b. P. Landry, R. Essick, and K. Chatziioannou, Phys. Rev. D101, 123007 (2020)
c. M. C. Miller et al., Astrophys. J. Lett. 887, L24 (2019)
d. B. Biswas et al., Phys. Rev. D 103, 103015 (2021)
Author Response
We strengthened the manuscript writing to better convey our approach. The revised version clarifies concerns raised by the reviewer, including a justification for the choice of the number of parameters used and a discussion on the MCMC convergence. We have used the SLy4 for low density and with more two polytropic, we could have some results that, we think, are not erroneous. The new version covers a detailed discussion about the methodology, addressing the 3 major points raised.
Reviewer 3 Report
In this work, the author employed a statistical approach to the nuclear equation of state relevant for neutron star calculation. The entire work is divided into two parts and there is no connection between them. In the first part, a Markov Chain Monte Carlo (MCMC) strategy is employed to explore the EOS domain with polytropic models and in the second part, they used a Bayesian power regression model to predict the credible region of EOS with the training performed on few known (65 eos) EOSs.
I found both parts of the work interesting but incomplete to the best of my knowledge. I have a few major difficulties in understanding the paper that is stopping me from passing the judgment to accept the article for publication in the present form.
The comments are as follows,
--------------
1st part of the work (Markov Chain Monte Carlo and Bayesian Inference):
Major:
1) As authors explore the space of the parameters $\Gamma_1$ and $\Gamma_2$ with the help of the Markov Chain Monte Carlo algorithm. I was completely confused to understand about their exact likelihood. What is the data they used in the likelihood? In order to calculate transition probability in the Metropolis algorithm one needs to calculate probability at each step. The parameter $\Gamma_1$ and $\Gamma_2$ will do a random walk but are guided by which likelihood? and which data? These need to explain in the manuscript.
2) "The set of parameters is assumed to be uncorrelated and normally distributed--"
The authors assumed normal distribution of parameters and those are uncorrelated. This is very hard to digest. Depending on the fit data considered in Bayesian Analysis will make the parameters correlated or uncorrelated. The authors should consider the flat distribution of parameters and make them evolve in MCMC, then see the effect of the fit data on the parameter distributions and also on corner plots.
3) In this work, they have generated EOS in MCMC and separately calculate the star properties, and commented on EOS with Recent Observations. I will rather recommend that it will be more justified if they consider the observational constraints also in part of the likelihood. I mean, NS maximum mass above 2 Msun, NICER radius, tidal deformability, etc. However, NS maximum mass and radius are the stringent ones compared to tidal deformability. The present boundary of tidal deformability constraints obtained from GW170817 is useless.
4) How they chose those values of the parameter $\Gamma1$=\{2.5,2.6,2.8,3.0\} and $\Gamma1$2=\{1.8,1.9,3.0,3.3,3.7\}?
The role of MCMC is not clear. One can generate the distribution along with 0.1 uncertainty even without MCMC. The role of inference is not clear in updating a prior belief.
Minor:
5) After Eq 10, f(x)=exp(-x) should be $f(\chi^2)=exp(-\chi^2/2)$
6) The total number of obtained EOS in each case should be mentioned in the text.
7) The star properties for a given mass need not be always normal distribution. So, instate of the mean one should calculate the median from CDF.
2nd part of the work (Bayesian Power Regression Model with Heteroscedastic errors):
Major:
This part I found more interesting. The authors have basically trained a Bayesian Power Regression Model with few known EOS and found the credible domains of EOS space.
1) Initially, all values of gamma are set to 1. But how are they after obtaining the final posterior?
2) Heteroscedastic errors capture the uncertainty and that has to be, it is a nice demonstration. But this needs to explore with a few robust tests. I think the author already plans to do this in the future. The 65 EOS considered here are actually what types of EOS? The reference given by them points out toward code by LIGO. What are the name of those EOSs? The author should refer to any article or explain it in brief in the text.
Minor:
3) If possible it should be tested on a larger EOS set. They could easily find a few recent articles giving EOS tables (also compOSE).
Author Response
Major point s
We have rewritten the entire article and the section regarding the MCMC, expanded and explained the hole of gamas as well as we do the distributions to address the points 1, 2, 3 and 4.
Major part 2
1 - We have not analyzed the posterior distribution for the parameters involved
here since we did not solve the TOV equations. We believe we need to
include this last step in the analysis before looking at the posteriors.
2- The reference is pointed to LIGO Lalsuite [95] in Chapt 5 - https://git.ligo.org/lscsoft/lalsuite/-/tree/master/lalsimulation/lib. Our goal in this work is to perform an initial investigation on the use of machine learning techniques and not to expand deeply on this analysis, for example, naming all the EoS and differences among them, the models and parametrization. We agree with the reviewer's comment. We will perform a deep investigation on machine learning models in a future work.
3 - We appreciate the reviewer's suggestion. We will take it into account for the upcoming studies.
Round 2
Reviewer 1 Report
I thank the authors for their response and careful consideration of my past comments. I believe that the quality of the paper has significantly improved with respect to the previous version, and I commend the authors for their effort. I believe that some major revisions are still necessary, however, and I urge them to consider my observations below.
Regarding Sec. 4:
1. For each sampled EOS (i.e., for each set of parameters {Gamma_1, Gamma_2} ) the authors now solve the TOV equations and compute the M(R) curve. I greatly appreciate this improvement. However, it is still unclear how the \chi^2 statistics that enters their likelihood is computed from the M(R) curve and the observational data. This is a key detail of the paper, and the authors should clarify this point and explicitly spell out the expression that they use for chi^2.
2. The authors state that "the set of parameters adjusted are assumed to be uncorrelated with uniform prior distributions". Why do the authors need to assume that the Gammas are uncorrelated? And what are the prior bounds that are being used for the three analyses?
3. When describing Fig. 4 the authors state that "The curves are expected to cross the region of the NICER observation [...] and at the same time reach massive pulsars with mass as high as M=2Msun". However, inspecting the figure, one immediately notices how not all M(R) curves satisfy these constraints. This seems to indicate that the MCMC sampling performed does not correctly implement the observational constraints, and that an EOS that e.g. does not reach a maximum mass of 2Msun is not being discarded (i.e., it has a likelihood larger than 0).
Note that this criticism is valid also for the models shown in Fig. 5
4. Finally, I appreciate that the authors attempted a sampling also in rho_2. The chains however do not seem to be converged, and the algorithm might have to be run for a larger number of iterations.
Round 3
